# BMP Signaling Interferes with Optic Chiasm Formation and Retinal Ganglion Cell Pathfinding in Zebrafish

**DOI:** 10.3390/ijms22094560

**Published:** 2021-04-27

**Authors:** Max D. Knickmeyer, Juan L. Mateo, Stephan Heermann

**Affiliations:** 1Department of Molecular Embryology, Institute of Anatomy and Cell Biology, Faculty of Medicine, University Freiburg, 79104 Freiburg, Germany; Max.Knickmeyer@anat.uni-freiburg.de; 2Faculty of Biology, University of Freiburg, Schaenzlestrasse 1, 79104 Freiburg, Germany; 3Departamento de Informática, Universidad de Oviedo, Jesús Arias de Velasco, 33005 Oviedo, Spain; mateojuan@uniovi.es

**Keywords:** chiasm, RGC axons, midline, BMP, shh, zebrafish

## Abstract

Decussation of axonal tracts is an important hallmark of vertebrate neuroanatomy resulting in one brain hemisphere controlling the contralateral side of the body and also computing the sensory information originating from that respective side. Here, we show that BMP interferes with optic chiasm formation and RGC pathfinding in zebrafish. Experimental induction of BMP4 at 15 hpf results in a complete ipsilateral projection of RGC axons and failure of commissural connections of the forebrain, in part as the result of an interaction with shh signaling, transcriptional regulation of midline guidance cues and an affected optic stalk morphogenesis. Experimental induction of BMP4 at 24 hpf, resulting in only a mild repression of forebrain shh ligand expression but in a broad expression of pax2a in the diencephalon, does not per se prevent RGC axons from crossing the midline. It nevertheless shows severe pathologies of RGC projections e.g., the fasciculation of RGC axons with the ipsilateral optic tract resulting in the innervation of one tectum by two eyes or the projection of RGC axons in the direction of the contralateral eye.

## 1. Introduction

Development of the vertebrate eye begins during late gastrulation in the prosencephalon, where an eye field is defined from which two optic vesicles evaginate [1,2,3]. Consecutive optic cup morphogenesis also is a dynamic process. Importantly, it requires an epithelial sheet migration over the distal rim of the optic cup integrating tissue from the lens-averted domain into the lens-facing domain [4,5,6,7]. Yet, optic stalk domains are also largely integrated into the optic cup via the ventral rim, which is the optic fissure [8,9]. These movements are facilitated by BMP antagonists [4,8]. The optic nerve head or optic disc, the area through which retinal ganglion cell axons exit the eye, is established at the proximal end of the optic fissure [1,2,10].

Retinal ganglion cells (RGCs) are the first type of cells differentiating from the retinal precursors at approximately 29 hpf in zebrafish. RGCs relay the visual information perceived by photoreceptors and pre-processed within the retina to visual centers of the brain [11]. The target areas of the RGC axons vary among different phyla and in humans are dominated by the geniculate pathway to the visual cortex. But projections to more basic brain regions like the optic tectum are also well described. In zebrafish, the optic tectum is the primary target area for RGC axons [12].

During evolution of bilateria, a concept of fiber decussation was established which entails for the visual system that information from the left visual field is transferred to the right side of the brain. In zebrafish, the eyes are positioned laterally and thus all RGC axons project to the contralateral side. On their way to the contralateral tectum, axons from both sides cross the midline and form the optic chiasm. In mice on the other hand, the eyes are positioned anterolaterally, resulting in a partially overlapping visual field. This results in a subtotal crossing of RGC axons, with some projecting ipsilaterally.

On the way to their target region, the RGC axons are directed to specific areas by distinct cues. At first, they are collected to project out of the eye through the optic disc, a process involving the localized expression of Netrin [13]. Consecutively, the axons extend towards the midline, where the decision is being made to cross or not to cross. The morphogen Shh was shown to be in control of axonal midline crossing of commissural axons in the spinal cord [14,15] but also affecting optic chiasm formation [16,17,18]. The findings suggest that Shh needs to be suppressed locally in order for the axons to cross the midline [19,20] while Pax2, expressed in the developing optic stalk, is guiding the axons towards the midline [19,21]. In mice, the Shh co-receptor Boc was shown to be expressed in the ipsilateral RGCs [22,23], while Shh was transported via RGC axons [24,25] and secreted locally at the chiasm, preventing Boc^+^ axons from crossing [26].

The regulation of axon crossing at the future optic chiasm also depends on multiple other factors varying considerably between species. A feature shared by mouse and zebrafish is the expression of Neuropilin-1 in contralaterally projecting RGCs. This membrane receptor interacts at the midline with VEGF in mouse and Semaphorin3D in zebrafish to promote axon crossing [11,27]. The lack of *sema3d* at the midline, but also ubiquitous overexpression, lead to the mis-projection of some RGCs to the ipsilateral side [28]. Another receptor important for RGC axon guidance is Roundabout-2 (Robo2), which mediates repellence from Slit proteins. Robo2 mutants feature multiple RGC guidance defects, including mis-projections at the midline [29]. In species with binocular vision, Ephrin-EphR signaling and repulsive guidance molecule a (RGMa) were found to be required for regulation of ipsilateral and contralateral projections [11,30].

Here, we show that BMP antagonism is essential for optic chiasm formation in zebrafish. We found that experimental induction of BMP4 at 15 hpf, affecting optic cup and optic stalk morphogenesis, results in a complete ipsilateral projection of RGC axons and failure of commissural connections in the forebrain. We identified reduced expression of Shh ligands and guidance molecules at the midline. Induction of BMP4 at 24 hpf, not affecting optic cup and stalk morphogenesis, resulted in different RGC projection defects, including the innervation of one tectum by two eyes (lambda phenotype) or projection into the contralateral eye. Removal of one optic cup after BMP4 induction resulted in contralateral projection of RGC axons, suggesting that the lambda phenotype is caused by erroneous fasciculation of the optic tracts.

## 2. Results

### 2.1. BMP Signaling-Induced Eye Malformations Do Not Affect Optic Nerve Formation But Disrupt Optic Chiasm Development

Defects of optic cup morphogenesis oftentimes affect formation of the optic fissure, resulting in morphogenetic coloboma [4,8,31]. The formation of the proximal end of the fissure, the presumptive optic nerve head, is also altered in these conditions. We used a model of optic cup morphogenesis defects through induction of *bmp4* expression in the zebrafish (*Danio rerio*) [8,31] to investigate the functional integrity of the presumptive optic nerve head.

We first examined the expression of optic stalk and optic nerve head markers *ntn1a* and *pax2a* in eyes of 30 hpf *tg(hsp70l:bmp4)* embryos after induction of *bmp4* at 15 hpf. We found that the expression of both was changed in the optic stalk (Figure 1A–D). While *ntn1a* expression was absent from the eye, *pax2a* was present. Since morphogenetic defects of the optic cup in this condition prevent proper formation of the optic fissure, the *pax2a*^+^ domain was located almost completely within the optic stalk (Figure 1C,D, also see [8]). With one ONH marker absent and another present, we wondered whether axon pathfinding towards the ONH would be impaired. Therefore, we studied a transgenic reporter line expressing membrane-bound GFP under the control of a regulatory element of *pou4f3* (also known as *brn3c*), which is active in a large subset of RGCs among other cell types [32]. RGC axons converged at the location of the distal optic stalk in both control and *tg(hsp70l:bmp4)* embryos after heat shock at 15 hpf, forming an optic nerve (Figure 1E–H). RGCs in the everted neuroretinal domains projected mostly directly medial, around the ventral rim of the optic cup towards the ONH (Figure 1H, asterisk), while some wrapped around the ventrotemporal margin of the irregular optic fissure (Figure 1G, arrow). RGC axons also remained confined to the retinal fiber layer and did not invade other retinal layers. Strikingly, however, the optic nerves of BMP-induced embryos did not cross the midline, but projected to the ipsilateral optic tecta instead (Figure 1I,J) (n = 69/81, 85.2%, as compared to n = 0/92, 0% in the control. Remaining embryos had ipsilateral projections into the anterior telencephalon or no projections.). In some cases, the optic nerve was divided into two fascicles, one rostral and the other caudal of the optic stalk, which due to morphogenetic defects [8] persists in BMP-induced embryos (Figure 1G,J, arrowheads). Both fascicles converged after passing the stalk while projecting dorsally towards the optic tectum (Figure 1J, left eye). Optic nerves in control embryos coursed through the diencephalon (as visualized by nuclear staining, H2B-RFP) and decussated at the midline, while those in BMP-induced embryos stayed within the space between eye and brain, heading immediately towards the ipsilateral optic tectum. The diencephalon appeared to extend further ventrally in BMP-induced embryos, without an opening allowing the optic nerves to pass through (Figure 1J, arrow). These results for the first time show an influence of BMP signaling on RGC projections. The observed trajectory of optic nerves suggests that in embryos with experimental induction of *bmp4*, RGC axons are unable to invade the diencephalon and approach the midline. This could be the result of ectopic presence of repellent guidance cues or absence of attractants, possibly because of a misspecification of midline tissue. Other axonal cues however appear to be intact, as optic nerve fasciculation and pathfinding to the optic tectum were unimpaired. Another possible explanation would be the absence of certain receptors from the axonal growth cones, rendering them insensitive to midline attraction.

In previous studies of ocular development, we observed expression of BMP antagonists in areas where BMP-sensitive processes occurred [4,8,31]. We therefore analyzed expression of BMP antagonists in the midline and eyes of wild type embryos. We found *follistatin a* (*fsta*) to be expressed in both the midline and the optic vesicles/optic cups between 16 and 24 hpf (Figure 1K–O). Expression of *noggin2* and *brorin* (*vwc2*) in the diencephalon was reported previously [33,34]. Additionally, BMP ligands are expressed at these developmental stages in the zebrafish brain, including in the olfactory placodes [35,36], the pineal gland [37] and the midbrain roofplate [38]. Together, this points towards the necessity of silencing of BMP signaling in this diencephalic domain to enable proper development. For a better understanding on how fast our model of *bmp4* induction could induce a transcriptional response, we crossed *tg(hsp70l:bmp4)* zebrafish to a BMP signaling reporter *tg(BRE-AAVmlp:eGFP)* [39]. We identified ectopic GFP signal as early as 4 h after heat shock, with strong widespread GFP labeling after 8 h (Appendix A). We also investigated whether physiological BMP signaling was required for the formation of contralateral retinotectal projections using the compound inhibitor DMH-2 and the inducible BMP antagonist *noggin3* in the transgenic line *tg(hsp70l:nog3)* [40,41]. A recurring result after DMH-2 treatment was the failure of one eye to produce an optic nerve. However, induction of *nog3* did not affect RGC projections and we did not observe any embryos in which RGC axons projected ipsilaterally under any treatment (Appendix A).

### 2.2. Ipsilateral Retinotectal Projections Are Part of a Larger Midline Defect in bmp4-Overexpressing Embryos

The phenotypes observed in embryos with induced expression of *bmp4* could be the result of a defect of midline patterning and guidance factors, or of an axonal defect of RGCs. We investigated zebrafish embryos expressing *bmp4* under the control of an *rx2* cis-regulatory element (*tg(Ola.rx2:bmp4)*, [4]), driving expression in retinal progenitor cells. These embryos present eye morphogenesis defects similar to those seen in early induced *tg(hsp70l:bmp4*) embryos [8]. We found contralateral retinotectal projections in all embryos analyzed at 4–5 dpf (n = 41), except one embryo with extremely close eye distance and very few, but exclusively ipsilateral RGC projections. Notably, in *tg(Ola.rx2:bmp4)* embryos with otherwise normal optic nerve trajectories, we observed an indentation in the optic nerves, where axons would head dorsally before turning and heading back ventromedially towards the optic chiasm. This indentation varied considerably in intensity (Figure 2A–C, arrows). 

The interpretation of these results requires caution, since the *rx2* cis-regulatory element is not exclusively active in retinal progenitors but to some degree also in the hypothalamus. Nevertheless, they suggest that the cause of the ipsilateral projection resulting from bmp4 induction unlikely lies within the axonal compartment, but is rather localized elsewhere, potentially at the midline. In order to test for a midline defect, we examined whether other forebrain axons were still able to cross the midline after *bmp4* induction. We studied the anterior commissure (AC) in the telencephalon and the postoptic commissure (POC) in the diencephalon, which are formed at 20–25 hpf. The outgrowth of these commissural fibers is preceded by the formation of a glial bridge that serves as a scaffold [42]. Hence, we visualized the commissures by immunohistochemistry for glial fibrillary acidic protein (gfap) and acetylated α-tubulin (AcTub) between 25 and 45 hpf (Figure 2D–K). At 25 hpf, the first axons of both AC and POC had crossed the midline and gfap^+^ glial cells formed bridge-like structures along the axonal paths in wild type embryos (3/3 embryos, Figure 2D, asterisks marking AC, arrowheads marking POC). In their *tg(hsp70l:bmp4)* siblings, however, glial bridges were disrupted and no axons crossed the midline (5/5 embryos, Figure 2E). This defect of midline structures recovered to a degree over the next hours. At 30 hpf, some POC axons had successfully crossed the midline in BMP-induced embryos, while the AC remained absent (Figure 2G). The first axons of RGCs appeared at the midline of wild type embryos at 35 hpf (Figure 2H, arrow). In some *tg(hsp70l:bmp4)* embryos, few AC axons had crossed the midline at this point. More POC fibers had grown across the midline, but the commissure appears disorganized and located more ventrally (Figure 2H,I). At the endpoint of our analysis at 45 hpf, the AC was formed in some embryos (3/5) but was much narrower than in controls (Figure 2J,K). The POC was more variable and appeared narrow and disorganized (3/5), or approximately normal (2/5) in different *tg(hsp70l:bmp4)* embryos. We concluded that *bmp4* overexpression at 15 hpf causes a strong midline defect of the forebrain.

### 2.3. Transcriptional Analysis after BMP Induction Reveals Misregulation of Numerous Axon Guidance Factors

Heat shock mediated induction of *bmp4* at 15 hpf resulted in exclusively ipsilateral retinotectal projections, presumably resulting from a midline defect. In order to uncover underlying transcriptional changes, we isolated total RNA from eyes and forebrain of *tg(hsp70l:bmp4)* embryos and wild type siblings at 6 h post heat shock. These samples were used for transcriptome analysis by microarray (Figure 3A). The results reflected the dorsalization of the eyes by BMP induction that we had observed previously [8], including upregulation of dorsal markers like *bambia/b* and *tbx3a*, and downregulation of ventral markers like *vax1/2*. We also found an upregulation of *fsta* [31] and other BMP antagonists like *nog1/2*. Interestingly, while most BMP-related genes were upregulated in response to *bmp4* overexpression (including *bmp7a* and *bmpr1aa/b*), we found strong negative regulation of *bmp7b* and *bmpr1ba* (Figure 3B). Among the differentially regulated genes, we identified 11 which are involved or potentially involved in axon guidance (Figure 3C). These include members of the Ephrin/Eph-receptor family (*efna1b*, *efnb3b*, *epha3*, *ephb2b*), Semaphorins/Plexins (*sema3ab*, *sema3d*, *plxnb1a*), Slits (*slit3*), Repulsive guidance molecules (RGMs, *rgmd*, *rgma*) and a leucine-rich repeat and immunoglobulin-domain containing protein (*lrig1*). We subsequently used whole mount in situ-hybridization (WMISH) to address the localization of the factors identified and to further analyze and verify the differential regulation observed in the microarray data. We confirmed the downregulation of *sema3ab*, *sema3d*, *rgmd*, *lrig1* and *ephb2b* (Figure 3D–O). Expression of *sema3ab*, even though it was strongly suppressed by bmp induction, was localized only to the dorsal optic cup in wild type embryos (Figure 3F,G) and was thus excluded from further analysis. The expression of the remaining four genes was not only regulated by *bmp4*, but also present in wild types in the diencephalon below the anterior optic recess where the optic chiasm is formed later during development. For *rgmd*, *lrig1* and *ephb2b*, the expression domains were extending from the diencephalic midline region into the optic stalk. *rgmd* is an RGM family member exclusively found in fish [43], which has not yet been functionally investigated. However, in chicken, RGMa was shown to be important for retinal axon guidance [30]. A role in axon guidance was described for murine Lrig2 [44], which is closely related to Lrig1. Furthermore, a receptor for leucine-rich repeats was shown to influence the laterality of RGC projections [45]. The role of *sema3d* in optic chiasm formation is well documented [27,28]. Both absence and ubiquitous expression result in ipsilateral misprojections of RGCs in zebrafish. However, only a part of RGCs projects ipsilaterally in these conditions and only 20% of *sema3d* mutants develop this phenotype. Therefore, downregulation of *sema3d* after *bmp4* induction cannot sufficiently explain the phenotype of exclusively ipsilateral projections. We analyzed the effect of nonsense mutations of *rgmd^sa44984^* and *lrig1^sa21873^* in homozygous embryos using injections of tracer dyes but did not observe phenotypes involving retinotectal projections. Furthermore, we employed a CRISPR/Cas9 protocol [46] to identify phenotypes in F0 embryos for *sema3d*, *ephb2b*, *rgmd* and *lrig1*. With this method, we also did not observe phenotypes related to RGC projections, both in single target and multiplexed injections. *rgmd* Crispants displayed necrotic tissue in the area of midbrain and hindbrain at 1 dpf, which was not observed in control embryos or Crispants for other genes. We concluded that the midline defect of *bmp4*-induced embryos was likely not caused solely by misexpression of these four guidance factors.

### 2.4. Sonic Hedgehog Ligand Expression Is Changed after bmp4 Induction

Previous research has shown that inhibition of sonic hedgehog (Shh) signaling or mutations in downstream targets (*gli1/2a*) leads to the formation of exclusively ipsilateral retinotectal projections [16,42]. Although we did not identify any transcriptional changes of Shh ligands, receptors or Gli transcription factors in our microarray analysis, we investigated the expression of Sonic hedgehog ligand genes *shha/b* in response to *bmp4* induction. In situ hybridization revealed expression of *shha* in ventral areas of the brain. Expression in the ventral diencephalon was specifically absent in *tg(hsp70l:bmp4)* embryos heat shocked at 15 hpf, while more caudally located expression was unaffected (Figure 4A–F). The paralog *shhb* showed expression in a limited domain below the optic recess in the ventral anterior diencephalon. This domain was absent in BMP-induced embryos (Figure 4G,H). We also analyzed midline expression of *slit* genes which were reported to cause defects of forebrain commissures in *gli1/2a* mutants [42]. Our analysis included WMISH against *slit1a* and *slit2* in embryos with induction of *bmp4* and control siblings. We found that midline expression next to the optic recess of *slit2*, but not *slit1a*, was inhibited by *bmp4* (Figure 4I–L). To follow up on these results, we analyzed how treatment with hedgehog signaling inhibitor Cyclopamine affects RGC projections in *tg(pou4f3:mGFP)* embryos. Previous data suggested that Shh inhibition starting at 15 hpf would result in ipsilateral projections [42], but was not clear on whether this referred to all or a subset of RGC axons. The treatment in our hands resulted both in embryos with exclusively ipsilateral projections as well as embryos with presumably normal contralateral retinotectal projections of one eye, while the axons of the other eye projected ipsilaterally, resulting in the innervation of only one tectum (Figure 4M–P). We termed this phenotype “lambda”, because of the similarity to the Greek small letter (λ). The lambda phenotype was reported to occur in a small subset of *dtr/gli1*, *yot/gli2a*, and *uml/boc* mutants [16]; however, the underlying cause remains elusive. Now that we were able to recapitulate the phenotype of *bmp4* overexpression by Shh inhibition, we wondered whether the downregulation of *sema3d*, *ephb2b*, *rgmd* and *lrig1* were resulting from the suppression of hedgehog signaling. Therefore, we analyzed the expression patterns of these genes in embryos treated with Cyclopamine. Interestingly, only downregulation of *sema3d* was detected while expression of the other three BMP-regulated genes was unaffected (Figure 4Q–X). In order to better understand the gene regulatory connections between BMP and Shh, we performed in-silico analysis of putative promotor regions of *shha/b*, *sema3d*, *rgmd*, *lrig1* and *ephb2b*, searching for binding motifs of BMP and Shh-dependent transcription factors (SMAD and GLI). We identified several SMAD-binding motifs in promotor sequences of both *shha* and *shhb* (Appendix A). Multiple SMAD-binding motifs were also present in the sequences analyzed for the other four genes, while only a single GLI-binding motif was identified in those, located upstream of *ephb2b* (Appendix A). Thus, surprisingly, no GLI-binding motif was found associated with *sema3d*, suggesting that either regulation of this gene by Shh is indirect or GLI-binding sites are located outside the sequence analyzed.

Earlier studies identified a link between Shh and Pax2 in the formation of proper RGC projections [19,21]. We therefore examined *pax2a* expression in *bmp4* induced embryos and in embryos treated with Cyclopamine. While at 21 hpf, expression was not noticeably changed in both conditions, the differences were striking at 42 hpf (Figure 5). In *tg(hsp70l:bmp4)* embryos, the *pax2a* expression domain was extended dorsally through the malformed optic stalk (Figure 5F, arrows). Notably, this part of the domain was located precisely in the aberrant path of optic nerves projecting to the ipsilateral optic tecta. In *tg(rx2:bmp4)* embryos, which also have a morphogenetic defect of the optic stalk, *pax2a* domain was changed in a similar, though less pronounced fashion (Appendix A). In contrast, embryos treated with Cyclopamine showed variable reduction of *pax2a* expression (Figure 5H–L), up to depletion of the domain (Figure 5H). This suggests that Shh signaling is required to maintain *pax2a* expression in the optic stalk and the ventral forebrain. We also screened the putative promotor sequences of *pax2a* in silico for SMAD and GLI binding motifs. Several such motifs were identified for both SMAD and GLI (Appendix A).

### 2.5. Late BMP Induction at 24 hpf Results in Different Projection Defects

In a previous project, we established that effects of *bmp4* overexpression on eye development differed substantially between heat shock timings before and after 24 hpf [31]. Early induction would produce defects of optic cup morphogenesis, while late induction would cause a fusion defect of the correctly formed optic fissure. Outgrowth of RGC axons does not occur until 28 hpf, thus allowing us to test whether the morphogenetically caused misplacement of *pax2a*^+^ cells was responsible for ipsilateral projections by induction of *bmp4* at 24 hpf. Late induction of *bmp4* did not result in exclusively ipsilateral projections, but produced different projection phenotypes (Figure 6A–D). One subset of embryos presented normal, exclusively contralateral retinotectal projections (n = 46/127, 36.2%). Others displayed the lambda phenotype with unilateral innervation of the optic tectum by both eyes (n = 50/127, 39.4%, no bias to left or right), while others showed no innervation of the tecta whatsoever but apparently retinoretinal projections, with both optic nerves coursing into each other at the midline (n = 15/127, 11.8%). In order to better characterize the aberrant projection phenotypes, we performed tracer dye injections into the eyes of fixed embryos at 3 dpf (Figure 6E–G). We confirmed that projections from both eyes reached the optic tectum in lambda embryos (Figure 6G). In the third group, retinoretinal projections were indeed formed, with few axons entering the fiber layer of the contralateral retina (Figure 6F). The size of the optic nerves decreased substantially between the midline and the contralateral eye, which suggests that most axons stalled in this area. The presence of aberrant retinoretinal projections was previously reported in *pax2a* mutants [21,47]. When investigating *pax2a* expression in embryos with late *bmp4* induction, we observed a striking change at the midline. Instead of being confined to a thin thread of cells ventral of the optic recess and absent from the midline, *pax2a* was expressed broadly in the anterior ventral forebrain, including expression close to the midline (Figure 6H–K). We further analyzed the expression patterns of *sema3d*, *rgmd*, *lrig1* and *ephb2b*. Transcription of *sema3d*, *lrig1* and *ephb2b* was reduced after late induction of *bmp4*, too, while *rgmd* expression was only mildly reduced (Appendix A).

Since RGC axons were able to cross the midline in embryos with late induction of *bmp4*, we expected that those embryos would not possess the midline defect we observed after early induction. We therefore examined AC and POC in *tg(hsp70l:bmp4)* embryos and controls subjected to heat shock at 24 hpf using immunohistochemistry. Overall, embryos with contralateral retinotectal projections also developed normal ACs and POCs (Appendix A). Individuals with irregular RGC projections possessed smaller, but otherwise normal POCs. In some cases, the anterior commissure was strongly reduced or interrupted at the midline (2/5). We also analyzed the expression pattern of *shha/shhb* after late induction of *bmp4*. At 30 hpf, expression of *shha* was not affected, while *shhb* expression displayed a similar reduction to after early induction of *bmp4* (Figure 6L–O). Cyclopamine treatment at 24 hpf resulted in mostly contralateral RGC projections (n = 37/48, 77.1%), but lambda and retinoretinal projections also occurred (8/48 lambda, 2 retinoretinal, 1 one-sided projection; Figure 6P). 

Taken together, we concluded that a late induction of *bmp4* results in a spectrum of mild to no midline defects, not acting via *shha* but potentially via *shhb* and likely involving the broadened *pax2a* domain.

### 2.6. RGC Axons Incorrectly Fasciculate with Contralateral Counterparts

Finally, we addressed the frequently appearing lambda phenotype both in embryos with late induction of *bmp4* as well as after Cyclopamine treatment. Since the phenotype does not form preferentially on either side of the brain, a genetic determination which tectum is innervated seemed unlikely. Instead, we hypothesized that pioneer axons from one eye arriving at the midline after their counterparts from the other eye already crossed could sometimes incorrectly fasciculate with those, therefore creating the lambda phenotype. In order to test this hypothesis, we induced *bmp4* expression in *tg(hsp:bmp4*, *pou4f3:mGFP)* embryos at 24 hpf and subsequently removed one optic cup. We analyzed the larvae at 3–4 dpf, assessing RGC projections by *pou4f3*-driven mGFP expression or DiI injection in *pou4f3:mGFP*-negative embryos. In total, we identified 43.2% lambda phenotypic embryos in transgenic embryos with two eyes (n = 19/44), but never observed ipsilateral projections from the remaining eye in treated embryos (n = 0/62, Figure 6Q). This clearly shows that ipsilateral projections in lambda phenotypic embryos form because of influence from RGC axons from the contralateral eye. Interestingly, removing this influence does not prevent formation of “retinoretinal” projections, as we still observed RGC axons projecting to the previous location of the eye removed in 9.7% of embryos (n = 6/62, 18.2% in controls, n = 8/44).

## 3. Discussion

We identified severe defects of RGC projections in zebrafish embryos with induced expression of *bmp4*. This is the first time to our knowledge that BMP signaling has been associated with RGC projections. When *bmp4* expression was induced at 15 hpf, embryos developed exclusively ipsilateral retinotectal RGC projections. Conversely, induction at 24 hpf resulted in different phenotypes in which midline-crossing was mostly intact, but other aspects of the retinotectal projection were not. The phenotype of complete conversion of retinotectal projections from contralateral to ipsilateral was previously reported in various zebrafish mutants, including *dtr/gli1*, *yot/gli2a*, *uml/boc*, *blw/ptch1*, *con/disp1*, *igu/dzip1* and *bel/lhx2b* [16,17,18,48,49,50,51]. In all these mutants, establishment of the midline is disrupted, oftentimes resulting in forebrain commissural defects in addition to the absence of the optic chiasm [42,48]. This was also the case in embryos with *bmp4* expression induced at 15 hpf, where the anterior and postoptic commissures (AC, POC) were disrupted. BMP signaling antagonist *brorin* was recently implicated in the formation of AC and POC [33]. Knockdown of *brorin (vwc2)* led to increased BMP signaling activity and disruption of commissure formation and axon guidance.

Apart from *bel/lhx2b*, the genes affected by abovementioned mutations are involved in Shh signaling. BMP and Shh signaling have antagonistic functions during multiple aspects of nervous system development [52,53]. We observed downregulation of *shha/b* after early induction of *bmp4*. Treatment of embryos with Shh pathway inhibitor Cyclopamine phenocopied the RGC projection and forebrain midline defects seen in the early and late BMP model when applied at 15 hpf (Figure 4M–P, [42]). Barresi and colleagues found, both in *yot/gli2a* mutants and wild type embryos treated with Cyclopamine, that the POC was more severely affected by disruption of Shh signaling than the AC. Conversely, in our BMP induction model, the AC was more severely affected than the POC, which recovered to a certain degree over time (Figure 2D–K). Here, *pax2a* likely plays a critical role, as it is required for proper POC development [21] and Shh signaling is required to maintain *pax2a* expression. The strong effect of *bmp4* on the AC could be a consequence of the inhibition of multiple axon guidance factors exclusively regulated by BMP like *rgmd* and *ephb2b*. Another factor might be the that the POC is located closer to the remaining *shha* expression domain in the midbrain, and is potentially exposed to a certain residual level of Shh. Interestingly, the optic chiasm next to the POC, did not recover over time, although it forms several hours later. Once the first retinal axons have misprojected ipsilaterally, subsequent axons likely use them as scaffold in a leader-follower interaction [54], preventing recovery. Our own and previous results suggest that axons of the POC do not project ipsilaterally, probably because they are only attracted by their target area after crossing the midline, unlike RGC axons.

We identified BMP-dependent genes encoding neuronal guidance cues (Figure 7) including *sema3d*, which was absent from the forebrain midline both after *bmp4* induction and Cyclopamine treatment. It was also downregulated in morphants of BMP antagonist *brorin* [33]. Misregulation of *sema3d* alone does not, however, result in defects as intense as we observed, but rather in guidance errors of some RGC axons, leading to mixed ipsilateral and contralateral projections [28]. The same was shown after inactivation of other parts of Semaphorin signaling, including *nrp1a*, *sema3e*, *adcy8*, *crmp4* and the sema-independent *islr2* [27,45,55,56]. For *rgmd*, *lrig1* and *ephb2b*, no mutant, KO or KD phenotype regarding RGC projections has been identified and they were not differentially regulated following Cyclopamine treatment. Thus, suppression of these genes does not sufficiently explain the projection phenotypes we observed. It should be noted that a plethora of axon guidance cues is expressed in the developing brain and a high degree of redundancy is to be expected. We therefore explicitly do not exclude possible roles for these genes. The importance of Slit and Robo genes for the optic chiasm is well documented [29,42,57]. The lack of *slit2* following *bmp4* induction very likely contributes to the midline defect in this condition. Overall, we propose that the midline defect can be attributed mostly to inhibition of Shh and downstream axon guidance genes.

While we observed many similarities between the phenotypes caused by early *bmp4* induction and Shh inhibition, there are some substantial differences. We already discussed some differences in gene expression regarding axon guidance cues, and effects on forebrain commissures. Furthermore, penetrance of the strongest ipsilateral retinotectal projection phenotype was higher in *bmp4*-induced embryos, and we did not observe RGC axons crossing the midline after early bmp4 induction. Conversely, Cyclopamine treatment resulted in ~65% embryos with RGC axons crossing the midline in some form. While it is possible that this phenotypic variability partially is due to variable efficacy of the inhibitor treatment, the lambda phenotype was reported to even occur in some individuals of Shh pathway mutants (*dtr/gli1*, *yot/gli2a*, *uml/boc*; [16]), further underlining that a crucial difference exists between the models of early *bmp4* induction and Shh inhibition.

We observed a striking difference regarding *pax2a* expression. This is interesting insofar as it was previously suggested that expression of Pax2 is repressed by BMP via repression of Vax1/2, while it is activated by Shh [52]. In accordance with these findings, we observed that Shh is required to maintain *pax2a* expression and that *vax1/2* expression is inhibited by *bmp4* (Figure 3A; [8]). In contrast however, *pax2a* was not downregulated after early *bmp4* induction, although the shape of the domain was changed due to morphogenetic defects. Moreover, induction of *bmp4* at 24 hpf led to an expansion of the forebrain *pax2a* domain, and multiple potential SMAD-binding motifs are present within the putative promotors of *pax2a*, suggesting that BMP can actually promote expression of *pax2a*. We propose that the negative effect on *pax2a* expression by Shh inhibition could be counteracted by early *bmp4* induction (Figure 7). Cells which express *pax2a* are precursors of glial cells required for proper guidance of the optic nerve [21]. We observed that ectopically located *pax2a* expressing cells were pointing towards the ipsilateral optic tectum, possibly enhancing the projection defect by misguiding RGC axons. In embryos expressing *bmp4* under the *rx2* promotor, RGC axons were able to cross the midline and this mislocalization was less pronounced. It could well explain, however, the small projection aberration of the optic nerve in these individuals (Figure 2A–C). In embryos treated with Cyclopamine, expression of *pax2a* was reduced unevenly between individuals, often remaining in proximal parts of the optic stalk and/or across the midline. Potentially, this variation could account for the phenotypic variability.

Besides the redirection of all RGC projections to the ipsilateral tectum, we also observed other RGC projection phenotypes, in which axons are able to cross the midline. They occurred as a result of both Cyclopamine treatment and induction of *bmp4* at 24 hpf. The most frequent phenotype was the innervation of one tectum by both eyes, while the other tectum received little to no RGC projections (“lambda”). This phenotype had been observed in a small subset of *dtr/gli1*, *yot/gli2a*, *uml/boc* and *bel/lhx2b* mutants [16,58]; however, the underlying cause was not investigated further. We were able to show for the first time by eye enucleation that this phenotype is the consequence of RGC axons erroneously fasciculating with axons from the contralateral eye. We can further conclude that in lambda projections, the axons first arriving at the midline always cross, and consecutively act as “pioneer” axons for RGC axons from the other eye. It is established that most axons during outgrowth follow other axons with the same target as themselves [59]. In the visual system, the first RGC axons to grow out have pioneer function and depend highly on guidance cues at the midline and along the optic tract to find their way through the optic chiasm and towards their targets. Later-born RGC axons mostly use these pioneer axons to grow along and are less dependent on guidance cues from other sources [54]. However, it is crucial for visual system development in zebrafish that these axons do not follow RGC axons originating from the contralateral eye. It was presented by Macdonald and colleagues that optic nerves are growing in discrete channels at the midline, which prevents interaction between them [21]. In zebrafish mutants of *pax2a*, these channels are no longer separated, resulting in RGC axons from both eyes coursing into each other. This causes various misprojections, including RGC axons projecting into the contralateral eye [21,47]. We observed these retinoretinal projections both after late induction of *bmp4* and treatment with Cyclopamine. Interestingly, we saw that *pax2a* was also absent or reduced in embryos treated with Cyclopamine, while the *pax2a* expression domain at the midline was expanded strongly after *bmp4* induction at 24 hpf. Yet, both treatments yield lambda or retinoretinal projections in some of the affected embryos. These results suggest that a precisely defined and localized *pax2a* expression domain is important to keep optic nerves separated and thereby promote contralateral RGC projections. Surprisingly, we still observed “retinoretinal” projections to the previous location of an eye after its removal, showing that this phenotype is not dependent on contralateral axons, unlike lambda. Thus, the optic nerve channels appear to be present even in absence of an optic nerve and accessible for RGC axons from the contralateral eye. Importantly, in *pax2a* mutants, optic nerves do not form complete retinoretinal or lambda projections, but instead form mixed projections with some axons to the contralateral eye or into the ipsilateral optic tract [21]. This suggests that additional defects are required to produce the phenotypes we observed. Possible explanations include the downregulation of axon guidance factors at the midline such as *sema3d*, and/or potential changes in the axonal compartment which promote fasciculation, such as increased expression of cell adhesion proteins like NCAM and L1CAM [60].

## 4. Materials and Methods

### 4.1. Zebrafish Care

Zebrafish were kept in accordance with local animal welfare law and with the permit 35-9185.64/1.1 from Regierungspräsidium Freiburg. The fish were maintained in a constant recirculating system at 28 °C on a 12 L:12 D cycle. Zebrafish embryos were grown in petri dishes in zebrafish medium, consisting of 0.3g/L sea salt in deionized water. If melanin-based pigmentation needed to be inhibited for downstream applications, embryos were grown in 0.2 mM phenylthiourea.

Transgenic lines used: tg(hsp70l:bmp4, myl7:eGFP) [31], tg(pou4f3:GAP-GFP) [32], tg(BRE-AAVmlp:eGFP, Xla.Eef1a1:H2B-mCherry) [39], tg(hsp70l:nog3) [40], tg(Ola.rx2:bmp4, myl7:eGFP) [4].

Mutant lines used: rgmd^sa44984^, lrig1^sa21873^ (obtained from the European Zebrafish Resource Center).

### 4.2. Heat Shock Procedures

For induction of heat-shock inducible transgenes, embryos were transferred to 1.5 mL reaction tubes and incubated at 37 °C in a heating block (Eppendorf Thermomixer, Hamburg, Germany). Embryos at 15 hpf are more sensitive and were therefore heat shocked for 20 min only, while Embryos aged 24 hpf were incubated for 1 h.

### 4.3. Analysis of Retinotectal Projections

The transgenic line *tg(pou4f3:mGFP)* was used to address retinotectal projections. Adult fish were crossed to AB wild type, *tg(hsp70l:bmp4*, *myl7:eGFP)*, *tg(Ola.rx2:bmp4*, *myl7:eGFP)*, or *tg(hsp70l:nog3)* individuals. Embryos were examined at 3–5 dpf using a stereomicroscope (Nikon SMZ18) with a light source for fluorescence. For confocal live imaging, embryos were microinjected at the 1-cell stage with 100 ng/µL H2B-RFP mRNA.

### 4.4. Confocal Laser Scanning Microscopy

Confocal images were recorded with an inverted TCS SP8 microscope (Leica, Wetzlar, Germany) with two internal hybrid detectors and a 40× long distance objective (water immersion). Live or fixed embryos were embedded in 1% low-melting agarose (Roth, Karlsruhe, Germany) in glass-bottom dishes (MatTek, Ashland, MA, USA). Live embryos were anaesthetized with MS-222 (Tricaine, Sigma-Aldrich, St. Louis, MO, USA) for imaging. Image stacks were recorded with a z-spacing of 3 µm, unless specified otherwise.

### 4.5. Image Processing

Images from microscopy were edited for presentation using ImageJ (Fiji) software [61].

### 4.6. In Situ Hybridization

Whole-mount ISH was performed according to [62]. For confocal imaging, embryos were stained using FastRed (Sigma-Aldrich) as the AP-substrate and DAPI as counterstaining. For probes used, refer to Appendix A.

### 4.7. Microarray Sample Preparation

Heterozygous *tg(hsp70l:bmp4)* embryos and wild type siblings from the same clutch were heat shocked at 15 hpf and sorted at the onset of *myl7:eGFP* expression at 21 hpf. Embryos were dissected by removing the yolk first, followed by dissection of forebrain and optic cups. Tissue from 10 embryos was pooled to create one sample. Three samples were collected as biological replicates per condition. RNA was isolated using an established phenol-chloroform-guanidinium thiocyanate extraction protocol [63].

### 4.8. Microarray Processing and Analysis

Samples were analyzed on GeneChip Zebrafish Gene 1.0 ST Array chips (902007, ThermoFisher, Madison, WI, USA) by the Genomics Lab of Dr. Dietmar Pfeifer (Klinik für Innere Medizin I, Universitätsklinikum Freiburg). The raw data from the CEL files were loaded and background corrected with the package *oligo* (version 1.46.0). Next the resulting matrices were quantile normalized using the packages *limma* (version 3.38.3). Each probe was linked with the corresponding gene ID based on the annotation in Ensembl version 95, probes matching to more than one gene were discarded and the abundance values were aggregated by gene using the median. The package *limma* was used to test for differential expression with the options *trend* and *robust*. The results were filtered by fold change (>1.15) and FDR (<0.05).

### 4.9. Lipophilic Tracer Dye Injection

Live or fixed embryos were embedded in 1.5% low-melting agarose (Roth) in glass-bottom dishes (MatTek). Live embryos were anaesthetized with MS-222 (Tricaine, Sigma-Aldrich). DiI or DiO (0.5% *w*/*v* in dimethylformamide, Merck) were injected into the retina using a microinjector (Eppendorf Femtojet 4i) and glass needles with a long taper.

### 4.10. Immunohistochemistry

For whole-mount staining, zebrafish were fixed at desired time points in 4% paraformaldehyde (PFA) in 1× PBS with 0.4% Tween-20 (1× PTW) overnight and rinsed with PTW. They were digested with Proteinase K (10 µg/mL) for an appropriate time and then permeabilized by incubation in acetone for 15 min at −20 °C, followed by rehydration in decreasing concentrations of methanol. Subsequently, embryos were incubated for 2–4 h in 1× PBDT (1× PBS, 1% BSA, 1% DMSO, 0.1% Triton X-100) with 10% normal goat serum for blocking. Primary antibodies were applied in blocking solution overnight at 4 °C. Afterwards, embryos were washed several times in 1× PBDT and then incubated with secondary antibody and 4 µg/mL 4’,6-diamidino-2-phenylindole (DAPI) in 1x PBDT overnight. Embryos were washed in 1x PTW and embedded for confocal imaging.

Mouse anti-acetylated α-tubulin (6-11B-1, Millipore, Burlington, MA, USA) 1:200Chicken anti-GFP (A10262, Invitrogen, Carlsbad, CA, USA) 1:500Rabbit anti-GFAP (Z0334, Agilent Dako, Santa Clara CA, USA) 1:200Goat anti-rabbit Alexa Fluor 647 (Cell Signaling Technologies, Danvers, MA, USA) 1:500Donkey anti-mouse Alexa Fluor 488 (R37114, ThermoFisher) 1:250Goat anti-rabbit Alexa Fluor 555 (A-21428, ThermoFisher) 1:250Anti-chicken Alexa Fluor 488 (Invitrogen) 1:500

### 4.11. Compound Inhibitor Treatment

Embryos were incubated in 100 µM Cyclopamine (Sigma-Aldrich) overnight starting at indicated timepoints. A stock solution of 10 mM in ethanol was diluted with zebrafish medium accordingly. Controls were treated with 1% ethanol.

Embryos were incubated with DMH2 [41] at indicated concentrations. A stock solution of 10 mM in dimethylsulfoxide (DMSO) was diluted with zebrafish medium to reach desired concentrations. Controls were treated with an equivalent amount of DMSO without an inhibitor. After treatment, embryos were rinsed twice with zebrafish medium.

### 4.12. Promotor Motif Search

For each or the genes analyzed, *pax2a*, *sema3d*, *rgmd*, *shha*, *shhb*, *ephb2b* and *lrig1*, their promotor region was defined as the window with respect to the transcription start site (TSS) of 2000 bases upstream and 500 bases downstream. In genes with several annotated TSSs, all of them were considered based on the Ensembl version 102. These sequences were scanned using FIMO [64] with the motifs from the Jaspar database MA0734.1 for GLI and MA1557.1 and MA0535.1 for SMADs. FIMO was run with a *p*-value threshold of 5 × 10^−4^. 

### 4.13. CRISPR/Cas9 F0 Analysis (Crispants)

Embryos in 1-cell stage were microinjected with 1 µM Cas9 protein (Alt-R S.p. Cas9 Nuclease V3, 1081059, Integrated DNA Technologies) and 1 µg/µL sgRNA mix as described by [46]. sgRNAs were designed using CCTop (http://crispr.cos.uni-heidelberg.de, accessed 1 November 2019) [65]. For sequences, please refer to Appendix A.

### 4.14. Enucleation of Optic Cups

Embryos derived from a cross of *tg(hsp70l:bmp4*, *myl7:eGFP)* x *tg(pou4f3:mGFP)* fish were heat shocked at 24 hpf, anaesthetized with MS-222 and embedded in 1% low-melting agarose subsequently. One optic cup was removed with a sharpened tungsten carbide needle (Fine Science Tools), then, embryos were carefully removed from agarose and grown before reaching 3–4 dpf. RGC projections in double transgenic embryos were analyzed using the *pou4f3:mGFP* transgene, while others were injected with DiI (see above).

## Figures and Tables

**Figure 1 ijms-22-04560-f001:**
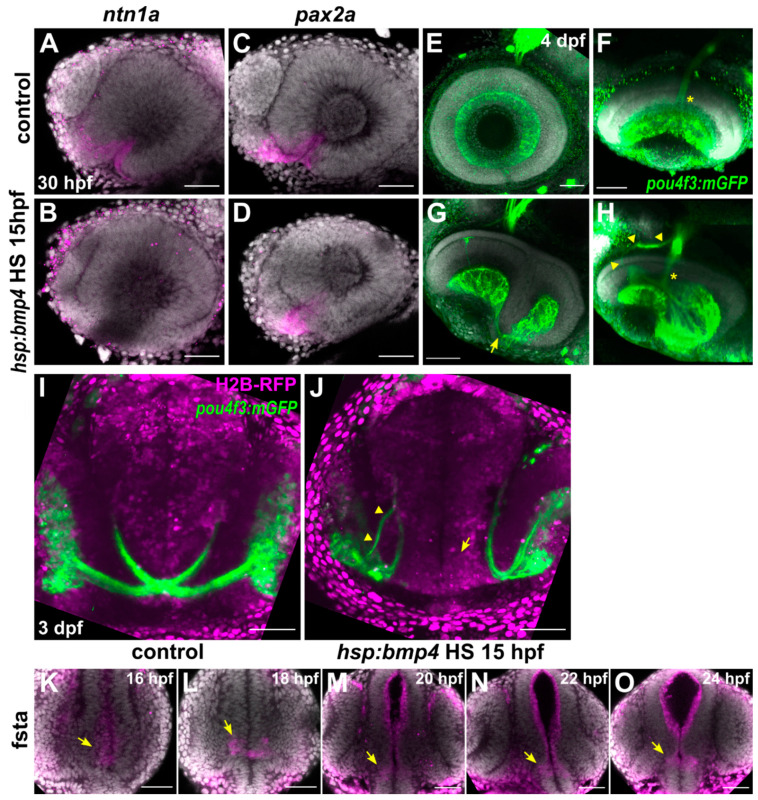
BMP signaling does not disrupt optic nerve head formation but prevents optic chiasm formation. (**A**–**D**) In situ hybridization of (**A**,**B**) *ntn1a* and (**C**,**D**) *pax2a* at 30 hpf in *tg(hsp70l:bmp4)* embryos and controls heat shocked at 15 hpf. DAPI counterstaining, sagittal view, nasal to the left. (**E**–**H**) Immunohistochemistry against GFP in *tg(pou4f3:mGFP*, *hsp70l:bmp4)* embryos and controls at 4 dpf. (**E**,**F**) Control eye, (**F**) tilted ventral projection. (**G**,**H**) Eye of an embryo with *bmp4* induced at 15 hpf. RGC axons wrapping around the rim from the everted temporal retinal domain (arrow). (**H**) Tilted 3D projection of the eye depicted in (**G**). ONH is intact (asterisk), there is an aberrant nasal branch of the optic nerve (arrowheads). DAPI counterstaining, sagittal view, nasal to the left. (**I**,**J**) Live imaging of *tg(pou4f3:mGFP*, *hsp70l:bmp4)* embryos and controls at 3 dpf. (**M**) Control embryo. (**N**) Transgenic embryo with *bmp4* induced at 15 hpf. RGCs project ipsilaterally. The right optic nerve possesses an additional nasal branch (arrowhead). There is no gap within the diencephalon (arrow). Transverse view. (**K**–**O**) In situ hybridization of *fsta*, midline expression (arrows) in the ventral prosencephalon of wild type embryos between 16–24 hpf. DAPI counterstaining, transverse view. All images are maximum intensity projections. Scale bars 50 µm.

**Figure 2 ijms-22-04560-f002:**
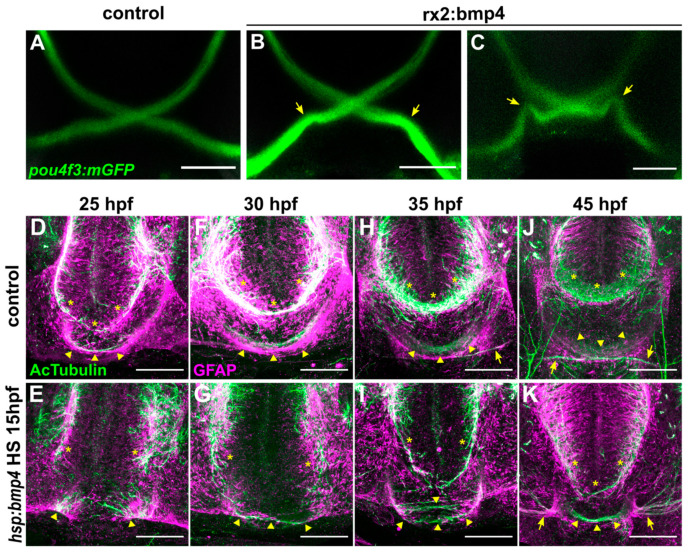
Ubiquitous bmp4 induction, but not eye specific bmp4 expression, results in a midline defect. (**A**–**C**) Immunohistochemistry against GFP in (**B**,**C**) *tg(pou4f3:mGFP*, *rx2:bmp4)* embryos and (**A**) controls at 4 dpf. Chiasms are intact, but optic nerve trajectories are irregular (arrows). (**D**–**K**) Immunohistochemistry against Ac-αTub and GFAP in (**E**,**G**,**I**,**K**) *tg(hsp70l:bmp4)* embryos and (**D**,**F**,**H**,**J**) controls at 25–45 hpf reveals the anterior (asterisks) and post-optic (arrowheads) commissures. Later stages include optic nerves (arrows). All images are maximum intensity projections, transverse view, scale bars 50 µm.

**Figure 3 ijms-22-04560-f003:**
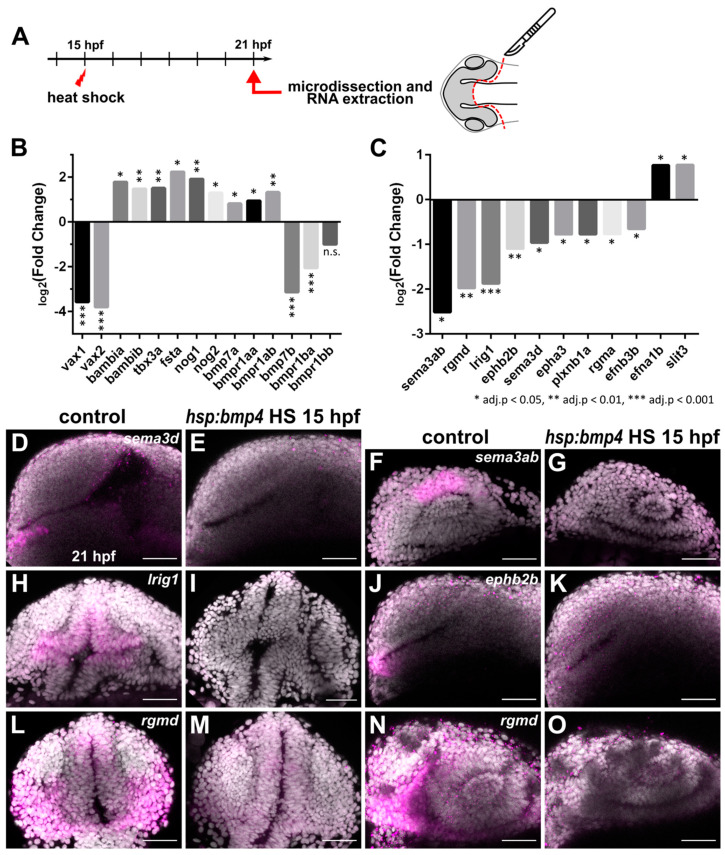
Transcriptomic analysis of forebrain tissue after *bmp4* overexpression reveals differential regulation of several axon guidance factors. (**A**) Scheme of sample collection for transcriptome analysis. (**B**,**C**) Expression changes of selected genes related to (**B**) optic cup patterning and BMP signaling, and (**C**) axon guidance, in *tg(hsp70l:bmp4)* embryos induced at 15 hpf, normalized to controls, as determined by microarray at 21 hpf. Expression level change displayed as log2(Fold change). * adj. *p* < 0.05, ** adj. *p* < 0.01, *** adj. *p* < 0.001. (**D**–**O**) In situ hybridization of genes related to axon guidance at 21 hpf in *tg(hsp70l:bmp4)* embryos and controls induced at 15 hpf. (**D**,**E**) *sema3d.* (**F**,**G**) *sema3ab.* (**H**,**I**) *lrig1.* (**J**,**K**) *ephb2b.* (**L**–**O**) *rgmd.* (**D**–**G**,**J**,**K**,**N**,**O**) Sagittal view, nasal to the left; (**H**,**I**,**L**,**M**) transverse view. All images are maximum intensity projections; counterstaining with DAPI, scale bar 50 µm.

**Figure 4 ijms-22-04560-f004:**
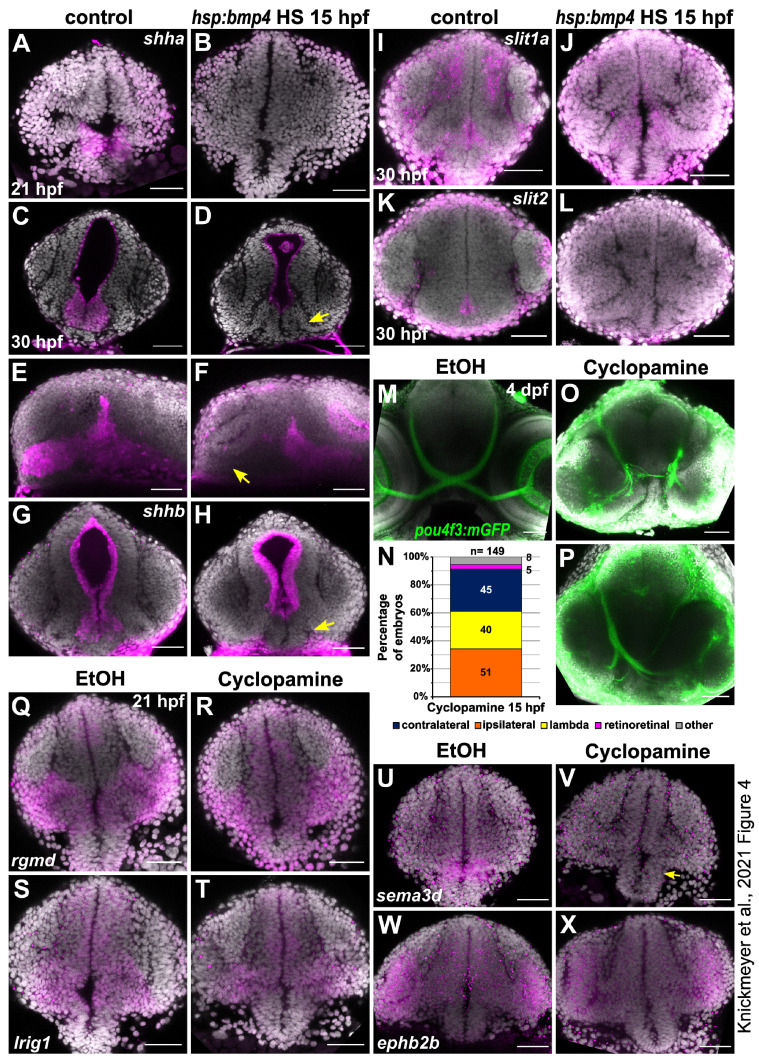
Sonic hedgehog expression is locally inhibited by bmp4 induction and inhibition of Shh signaling results in defective RGC projections. (**A**–**L**) In situ hybridization in *tg(hsp70l:bmp4)* embryos and controls heat shocked at 15 hpf. (**A**,**B**) *shha* expression at 21 hpf in the diencephalon is inhibited by *bmp4*. (**C**–**F**) *shha* expression at 30 hpf in the anterior domain of the diencephalon (arrow) is inhibited by *bmp4*. (**G**,**H**) *shhb* expression at 30 hpf in the diencephalon (arrow) is inhibited by *bmp4*. (**I**,**J**) midline *slit1a* expression at 30 hpf is not changed by *bmp4*. (**K**,**L**) *slit2* expression at 30 hpf is inhibited by *bmp4.* (**M**–**P**) RGC projection phenotypes in *tg(pou4f3:mGFP)* embryos treated with Cyclopamine. (**M**) Ethanol control. (**N**) Chart showing the distribution of RGC phenotypes in embryos treated with 100 µM Cyclopamine. Controls had exclusively contralateral projections (n = 61). (**O**) Ipsilateral RGC projections and (**P**) unilateral innervation of an optic tectum (“lambda”) in treated embryos. (**Q**–**X**) In situ hybridization in (**R**,**T**,**V**,**X**) embryos treated with 100 µM Cyclopamine and (**Q**,**S**,**U**,**W**) controls treated with ethanol. (**Q**,**R**) Expression of *rgmd*, (**S**,**T**) *lrig1.* (**U**,**V**) Expression of *sema3d* in the anterior diencephalon is sensitive to Cyclopamine treatment (arrow). (**W**,**X**) Expression of *ephb2b*. All images are maximum intensity projections, DAPI counterstaining, transverse view (except **E**,**F**, sagittal view). Scale bars 50 µm.

**Figure 5 ijms-22-04560-f005:**
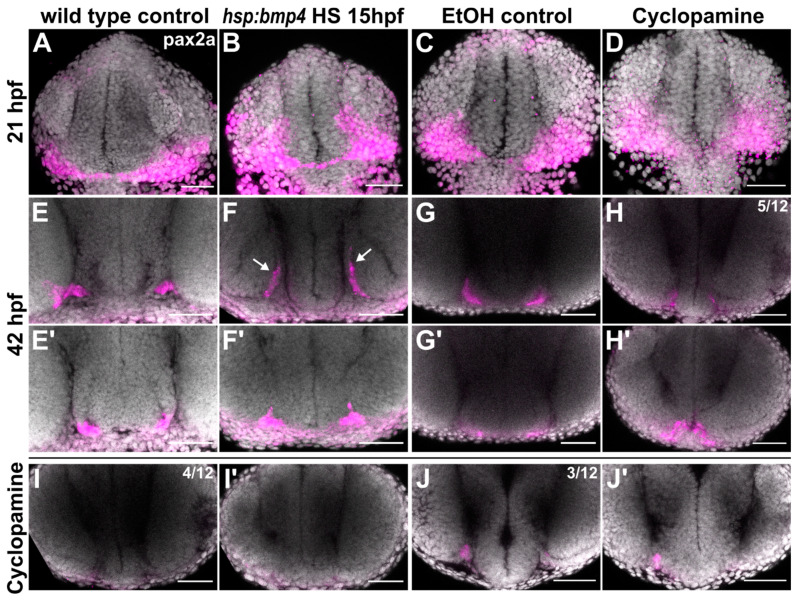
*pax2a* expression domains are differentially altered in *bmp4* induction and Shh inhibition conditions. (**A**–**J**) In situ hybridization of *pax2a*. (**A**–**D**) 21 hpf in (**A**) control and (**B**) *tg(hsp70l:bmp4)* embryos after heat shock at 15 hpf; (**C**) ethanol control and (**D**) Cyclopamine-treated embryos. (**E**–**J**) 42 hpf in (**E**) control and (**F**) *tg(hsp70l:bmp4)* embryos after heat shock at 15 hpf. Part of the *pax2a* domain extends dorsally (arrows). (**G**) Ethanol control and (**H**–**J**) Cyclopamine-treated embryos with different expression patterns (n = 12); (**E’**–**J’**) show a more anterior region in the same embryos as (**E**–**J**). All images are maximum intensity projections, DAPI counterstaining, transverse view, scale bars 50 µm.

**Figure 6 ijms-22-04560-f006:**
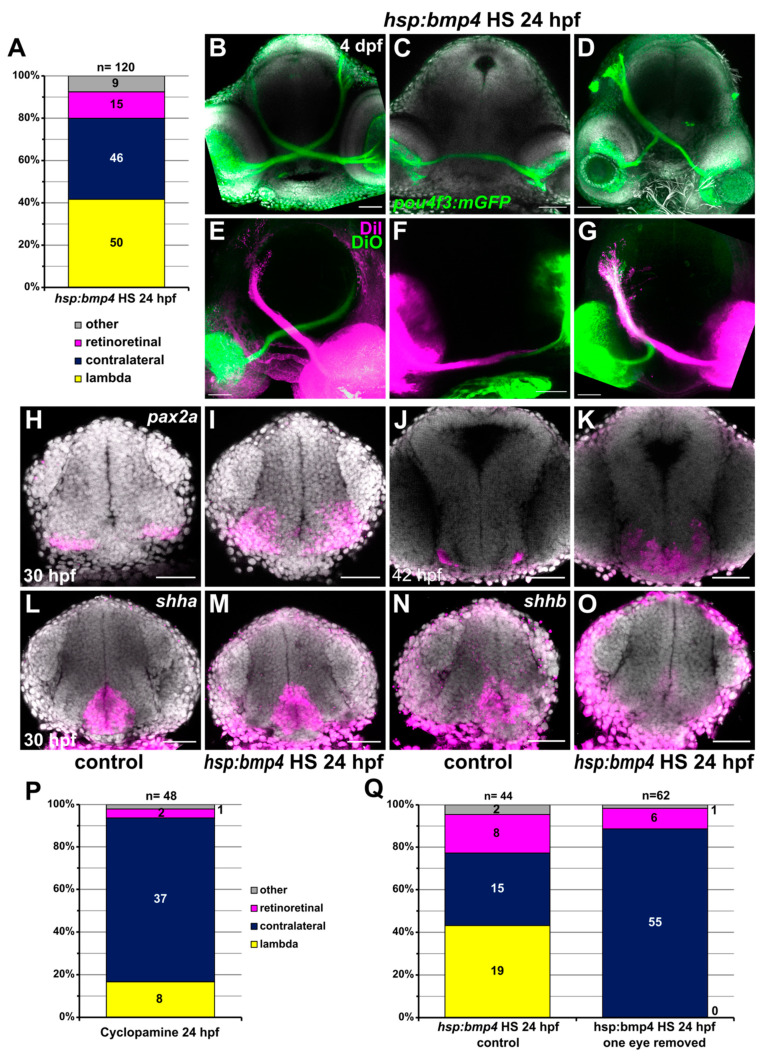
Late induction of *bmp4* at 24 hpf results in different RGC projection defects. (**A**) Chart showing the distribution of phenotypes in *tg(hsp70l:bmp4)* embryos after heat shock at 24 hpf. Controls had exclusively contralateral projections (n = 78). (**B**–**G**) RGC projection phenotypes in *tg(hsp70l:bmp4)* embryos after heat shock at 24 hpf. (**B**–**D**) Immunohistochemistry against GFP in *tg(pou4f3:mGFP)*, DAPI counterstaining, 4 dpf. (**E**–**G**) Injection of DiI/DiO, 3 dpf. (**B**,**E**) contralateral phenotype, (**C**,**F**) retinoretinal phenotype, (**D**,**G**) lambda phenotype. (**H**–**O**) In situ hybridization of in (**I**,**K**,**M**,**O**) *tg(hsp70l:bmp4)* embryos and (**H**,**J**,**L**,**N**) controls after heat shock at 24 hpf. (**H**,**I**) *pax2a*, 30 hpf; (**J**,**K**) *pax2a*, 42 hpf; (**L**,**M**) *shha*, 30 hpf; (**N**,**O**) *shhb*, 30 hpf, DAPI counterstaining. (**P**) Chart showing the distribution of phenotypes in embryos treated with 100 µM Cyclopamine at 24 hpf. Controls had exclusively contralateral projections (n = 30). (**Q**) Chart showing the distribution of phenotypes in *tg(hsp70l:bmp4)* embryos after heat shock at 24 hpf and subsequent removal of one eye, and in controls. All images are maximum intensity projections, transverse view, scale bars 50 µm.

**Figure 7 ijms-22-04560-f007:**
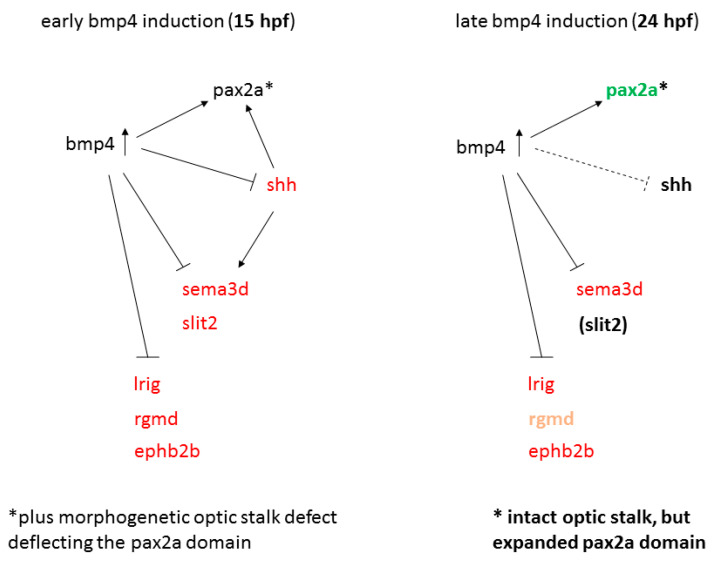
Model of gene regulation in early and late *bmp4* induction models. Red and green indicate reduced and increased expression, respectively.

## Data Availability

Expression data are available under: https://www.ebi.ac.uk/arrayexpress/experiments/E-MTAB-10085, 26 April 2021.

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
