# Peer review of "BMP Signaling Interferes with Optic Chiasm Formation and Retinal Ganglion Cell Pathfinding in Zebrafish"

_ijms, 2021, doi:10.3390/ijms22094560_

Round 1
Reviewer 1 Report
The manuscript is interesting and well written. The material and methods are sound and the imagines are of high quality.
The authors might also analyze using confocal microscopy the details of the morphological changes and in other aspects of retinal development that might occur also in other cellular types of the retina including the photoreceptor layer.
Author Response
Dear Ms. Zurini Fu and dear Editors at the IJMS,
at first, we want to thank the Editorial Office and the Referees for the constructive comments on our work. We are happy that the overall quality of data was considered to be good. Please find below our answers to the Referees comments and questions, point by point. We are happy to resubmit our revised work and are looking forward to hearing from the editorial office.
Best Regards
Stephan Heermann
Ad Reviewer 1:
The manuscript is interesting and well written. The material and methods are sound and the imagines are of high quality.
The authors might also analyze using confocal microscopy the details of the morphological changes and in other aspects of retinal development that might occur also in other cellular types of the retina including the photoreceptor layer.
We want to thank the referee for their kind evaluation. We have previously investigated and documented the morphological changes in optic cup development after induction of bmp4 in great detail (Eckert et al. 2019, Reference 8 of this manuscript). The referee points out another very interesting aspect of eye development in differentiation of the retinal cell types. We have not specifically addressed photoreceptor development in the context of this manuscript as it is not affecting RGCs and the retinotectal projection. However, in a previous study it is shown that retinal differentiation is occurring at least to some degree under the influence of high levels of BMP4 (Heermann et al., 2015). In these analyses, BMP4 was expressed using a rx2 cis regulatory element (tg(rx2:bmp4)). The morphogenetic defects of the optic cup were similar to those of tg(hsp70l:bmp4) embryos induced at 15-17 hpf (please see also Eckert et al., 2019). Transgenic reporters of vsx1 and vsx2 were used to show early differentiation of retinal progenitors (Heermann et al. 2015, Fig. 6 – Supplement 1).
Reviewer 2 Report
In the present study, the authors have investigated the role of BMP signaling in the formation of the optic chiasm and retinal ganglion cell pathfinding in zebrafish. They found that over expression of BMP at different times post fertilization resulted in either complete ipsilateral projection of RGC axons (15 hpf) or in milder phenotypes (24 hpf). They further found that inhibition of Shh pathway mimics these phenotypes. They also identified BMP-dependent genes that are involved in neuronal guidance and how the midline is malformed in these mutants. The phenotypes observed after overexpression of BMP and inhibition of Shh signaling, however, were not exactly same. They also showed that Shh is required to maintain the expression of pax2a. In addition to the redirection phenotypes, the authors also observed "lambda" phenotypes, which they explained is the result of an error in RGC axons fasciculation.
The manuscript is written nicely. The methods are explained well.
The introduction, results and discussion sections are also very well written. The figures are clear along with the legends.
I only have a few questions -
1) When is BMP expressed in the brain and where all, especially in the context of their experiments? Could they add some in situs showing the expression pattern or cite some previous study?
2) Line - 490-491 - Is the midline defect happening due to misguided RGC axons or the axons getting misguided due to a malformed midline?
3) Why is the size so reduced in figure 5B?
Author Response
Dear Ms. Zurini Fu and dear Editors at the IJMS,
at first, we want to thank the Editorial Office and the Referees for the constructive comments on our work. We are happy that the overall quality of data was considered to be good. Please find below our answers to the Referees comments and questions, point by point. We are happy to resubmit our revised work and are looking forward to hearing from the editorial office.
Best Regards
Stephan Heermann
Ad Reviewer 2:
In the present study, the authors have investigated the role of BMP signaling in the formation of the optic chiasm and retinal ganglion cell pathfinding in zebrafish. They found that over expression of BMP at different times post fertilization resulted in either complete ipsilateral projection of RGC axons (15 hpf) or in milder phenotypes (24 hpf). They further found that inhibition of Shh pathway mimics these phenotypes. They also identified BMP-dependent genes that are involved in neuronal guidance and how the midline is malformed in these mutants. The phenotypes observed after overexpression of BMP and inhibition of Shh signaling, however, were not exactly same. They also showed that Shh is required to maintain the expression of pax2a. In addition to the redirection phenotypes, the authors also observed "lambda" phenotypes, which they explained is the result of an error in RGC axons fasciculation.
The manuscript is written nicely. The methods are explained well.
The introduction, results and discussion sections are also very well written. The figures are clear along with the legends.
We are grateful for the referee’s kind evaluation of our manuscript.
I only have a few questions -
1) When is BMP expressed in the brain and where all, especially in the context of their experiments? Could they add some in situs showing the expression pattern or cite some previous study?
We thank the referee for their comment. Veien et al., 2008 (Fig. 2) and French et al., 2009 (Fig. 3) show expression of bmp4 in the brain at the time in question. Schmid et al., 2000 (Fig. 4) present expression of bmp7a in the developing pineal gland. Active BMP signalling in this area can also be seen in our BMP reporter data (this study, Fig. S1). bmp5 is expressed in the midbrain roofplate according to Miyake et al., 2013 (Fig. 5). We reference these studies in the revised version of the manuscript.
2) Line - 490-491 - Is the midline defect happening due to misguided RGC axons or the axons getting misguided due to a malformed midline?
The referee is correct to point out imprecise language in these lines, where we used “midline defect” when talking about the RGC projection defect. We clarified this sentence in the revised version of the manuscript. It now reads: “We observed that ectopically located pax2a expressing cells were pointing towards the ipsilateral optic tectum, possibly enhancing the projection defect by misguiding RGC axons.”
3) Why is the size so reduced in figure 5B?
We thank the referee for pointing this out. Embryos induced with bmp4 can sometimes be slightly smaller than controls, however this was not the case here. The stark difference in size in Figure 5B was due to the usage of different focal planes (more anterior than in 5A, C-D). We replaced the image using the corresponding focal planes.